# Alteration, Reduction and Taste Loss: Main Causes and Potential Implications on Dietary Habits

**DOI:** 10.3390/nu12113284

**Published:** 2020-10-27

**Authors:** Davide Risso, Dennis Drayna, Gabriella Morini

**Affiliations:** 1Ferrero Group, Soremartec Italia Srl, 12051 Alba, CN, Italy; 2National Institute on Deafness and Other Communication Disorders, NIH, Bethesda, MD 20892, USA; drayna@nidcd.nih.gov; 3University of Gastronomic Sciences, Piazza Vittorio Emanuele 9, Bra, 12042 Pollenzo, CN, Italy; g.morini@unisg.it

**Keywords:** taste, taste impairments, dysgeusia, hypogeusia, COVID-19, dietary habits

## Abstract

Our sense of taste arises from the sensory information generated after compounds in the oral cavity and oropharynx activate taste receptor cells situated on taste buds. This produces the perception of sweet, bitter, salty, sour, or umami stimuli, depending on the chemical nature of the tastant. Taste impairments (dysgeusia) are alterations of this normal gustatory functioning that may result in complete taste losses (ageusia), partial reductions (hypogeusia), or over-acuteness of the sense of taste (hypergeusia). Taste impairments are not life-threatening conditions, but they can cause sufficient discomfort and lead to appetite loss and changes in eating habits, with possible effects on health. Determinants of such alterations are multiple and consist of both genetic and environmental factors, including aging, exposure to chemicals, drugs, trauma, high alcohol consumption, cigarette smoking, poor oral health, malnutrition, and viral upper respiratory infections including influenza. Disturbances or loss of smell, taste, and chemesthesis have also emerged as predominant neurological symptoms of infection by the recent Coronavirus disease 2019 (COVID-19), caused by Severe Acute Respiratory Syndrome Coronavirus strain 2 (SARS-CoV-2), as well as by previous both endemic and pandemic coronaviruses such as Middle East Respiratory Syndrome Coronavirus (MERS-CoV) and SARS-CoV. This review is focused on the main causes of alteration, reduction, and loss of taste and their potential repercussion on dietary habits and health, with a special focus on the recently developed hypotheses regarding the mechanisms through which SARS-CoV-2 might alter taste perception.

## 1. Introduction

The senses allow us to detect signals from the environment, to codify and transmit them to the central nervous system, and to provide data for perception. Humans are chemoheterotrophic organisms that ingest and digest a wide range of chemical compounds. This process is mediated by taste and smell, the two of our senses known as chemical senses and, therefore, we talk about chemoreception. Smell and taste respond to chemical signals: small concentrations of volatile substances for smell, which allows remote sensing, and molecules dissolved in the saliva for taste, which provides proximal sensing. Chemesthesis also contributes to chemoreception. Hotness, pungency, and coolness are chemically induced sensations involving sensors that can be activated both by temperature (physical stimulus) and by specific substances contained in some foods [1]. These highly specialized, functionally integrated perception systems evolved to allow the identification of nutrients while avoiding potential toxic compounds and indigestible materials, and they participate in the control of motivational processes that guide dietary selection, driving food choices that nourish and maintain health [2].

Although taste and flavor are commonly conflated and used synonymously [3], flavor is the “complex combination of the olfactory, gustatory, and trigeminal sensations perceived during tasting”, as defined by the International Standards Organization [4]. In the human brain, the chemical senses perceptual areas are closely linked to and integrated with those for other sensory systems, with homeostatic and visceral processes, memory, emotion, and language [5,6,7]. Therefore, flavor perception is both innate and acquired (and has thus been suggested to be a form of intelligence), making it one the most complex of human behaviors, but the one on which feeding behavior and food choices strongly depend [8].

In this paper, we will primarily deal with taste, which we define as its scientific meaning that encompasses the basic tastes of sweet, umami, salty, bitter, and sour, to which additional qualities such as fatty, metallic, and others may be added [9].

The human taste system has developed over millions of years of evolution, which were characterized by a scarcity of food and, therefore, by difficulty in acquiring nutrients. The resulting taste code that humans now possess is quite simple: sweet substances are generally appreciated and include carbohydrates, an important source of energy; amino acids, some of which provide an umami taste (very pleasant in humans), are the constituents of proteins and are fundamental for our metabolism since they are the only source of nitrogen. In addition, some amino acids cannot be synthesized by humans and are thus essential. The appreciation of the salty taste (actually the taste associated with the presence of Na^+^ ions) has developed in mammals during their evolution far from the sea, to maintain the concentration of sodium ions (which are uncommon in nature) at necessary levels. Sour taste, which can be a sign of unripe fruits and spoiled food due to uncontrolled fermentation, can be tolerated and appreciated only to a certain extent. Even bitter substances can be accepted, but only in low concentrations and are never desirable to babies. Most bitter tastants are produced by plants, which have developed the strategy of accumulating bitter or irritating secondary metabolites (such as polyphenols, flavonoids, isoflavones, terpenes, and glucosinolates) to defend themselves against herbivores and pathogens. This does not mean that what is bitter is always toxic and, therefore, totally rejected. On the contrary, many of these bitter or chemesthetic secondary metabolites, once ingested, have positive actions on health, but humans learn to appreciate them only as they mature.

## 2. How We Taste

Tastants are detected by specific transmembrane receptors located at the apex of the taste receptor cells (TRCs). Taste receptors are capable of recognizing the molecules contained in potential food without the chemical entering the cell, and they reside within structures known as taste buds, which are distributed in the various papillae of the tongue and the soft palate. The circumvallate papillae are found at the rear of the tongue. In humans, they contain about one thousand taste buds. The foliate papillae are present at the posterior lateral edges of the tongue and contain roughly a dozen taste buds, while the fungiform papillae contain only a few taste buds and are found primarily on the tip and, to a lesser degree, up to 2/3 the length of the tongue. The filiform papillae (the most numerous) do not contain taste buds but are involved in tactile perceptions [10]. Studies in the past several years using molecular and functional data have shown that the various papillae are not highly selective for a specific taste, and therefore the old assignment of fundamental tastes to specific places on the tongue has been abandoned, although regional differences in responsiveness of some tastes have been recently reported [11,12,13].

There are two types of transmembrane receptors relevant for taste: ion channels, which mediate sour (H^+^) and salty (Na^+^) tastes, and G-protein-coupled receptors (GPCRs), which mediate sweet, umami, and bitter tastes, as well described in previous reviews [14,15].

Acidic taste has been recently discovered to be mediated by the ion channel Otopetrin-1, a proton-selective channel previously known to be involved in the sensation of gravity in the vestibular system [16]. Sour sensing is also mediated by the transient receptor potential cation channel subfamily V member 1 (TRPV1, also known as the capsaicin receptor), which is a sensor involved in the perception of irritants and other potentially tissue-damaging stimuli, including both exogenous compounds, such as capsaicin in chili peppers, and endogenous compounds, such as those produced during inflammation [17]. Inflammation and tissue damage produce a significant local increase in proton concentration, which is able to excite TRPV1 [18].

Na^+^, the main chemical ion responsible for salty taste, is important in many physiological processes. Low-to-medium concentrations of table salt (NaCl, the common sodium-containing chemical we use to season foods) are perceived as pleasant and appetitive, while high concentrations are aversive. There is still debate about the receptor(s) involved in Na^+^ perception, with the epithelial sodium channel (ENaC) being a leading candidate for this function [19].

There are more information and scientific certainty about the nature of the receptors for sweet, umami, and bitter tastes, which belong to the GPCR class. Given that the overall number of sweet, umami, and bitter taste receptors genes and their known variants is large, and that many reviews have already covered this topic [14,15,20], we here provide a short introduction to this topic to provide a general background for the reader and provide context on taste disturbances in the current Coronavirus disease 2019 (COVID-19) pandemic.

Naturally occurring sweet substances, although structurally diverse, are not many, and the great majority can be identified as small molecules, mainly mono- or poly-hydroxylated, such as carbohydrates and polyols, and some amino acids and peptides. Although large molecules such as sweet proteins like brazzein and thaumatin also taste sweet, they are rarities of the natural world that activate sweet receptors [21]. Many synthetic compounds and their derivatives also taste sweet, such as saccharin, cyclamate, and sucralose to name a few, while others act as sweetness modulators and sweetener enhancers [22]. This chemical diversity of sweet compounds has long fueled a discussion of whether there was a single receptor capable of binding all these compounds. However, in 2001, several independent groups simultaneously identified the sweet taste receptor as the heterodimer of taste receptor type 1 members 2 and 3, (T1R2/T1R3). This receptor responds to all sweet substances tested to date due to several binding sites able to sense the various ligands, as well as the substances capable of inhibiting the perception of sweetness, such as lactisole [23]. Our understanding of the sweet taste receptor at the molecular level has also allowed us to speculate on another important phenomenon, the synergism exercised by certain sweet compounds when used in mixtures. It now seems likely that the simultaneous occupation of two different sites in the receptor causes the sweet taste response to be enhanced by such mixtures [24].

The umami receptor is also a heterodimer and it shares a subunit with the sweet receptor, thus is composed of one copy each of T1R1 and T1R3. There seem to be only a few compounds capable of stimulating this receptor in humans, with L-glutamate exhibiting the strongest effect. Although the purine nucleotides such as inosine-5′-monophosphate (IMP) and guanosine-5′-monophosphate (GMP) evoke a modest umami taste on their own, they display considerable synergy with L-glutamate. This, in fact, was discovered and exploited by the food industry to formulate flavor enhancers well before identifying the specific umami receptor and understanding its mechanism of action [25].

Unlike sweet and umami tastes, which are able to identify a limited number of molecules, bitter taste performs the role of controlling the ingestion of a very large number of structurally different compounds, mainly from plants which have developed the strategy of accumulating toxic or irritating secondary metabolites (including polyphenols, flavonoids, isoflavones, terpenes, and glucosinolates) to defend themselves against herbivores and pathogens (Figure 1).

To respond to such a broad range of different compounds, mammals have evolved a large array of bitter receptors. Humans possess 25 bitter taste receptors (encoded by taste receptor type 2, -TAS2R- genes) that are considered to work as monomers, although some authors have suggested they can also work as homo and heterodimers as a way to further enlarge the spectrum of compounds that they can recognize. Some bitter receptors are able to sense (bind) many substances, some of which have very different structures, while others appear to bind only a few compounds, all of the similar structure. In addition, some bitter compounds are able to activate more than one bitter receptor (usually at very different concentrations) [26].

Until a few years ago, taste sensors were generally believed to reside only in the oral cavity. Recently, gustatory chemoreceptors (in particular sweet, umami, and bitter receptors) have been identified in other locations in the body, including the gastrointestinal tract (GI) and the respiratory system. Thus, taste and taste receptors have to be considered to be a broad chemosensory system that recognizes both exogenous (i.e., from food) and endogenous agonists (i.e., metabolites and compounds produced by the body and its microbiota) mediating systemic response [27]. The activation and modulation of taste receptors within the GI system by the same compounds that provide taste present in food may have very different functions, ranging from absorption to the metabolization of nutrients, the release of satiety hormones, the modulation of glucose transportation, gastric emptying, and gut motility regulation. These discoveries have now made it clear that taste receptors and their polymorphisms are associated with human disorders that go far beyond the oral cavity, a fact to be considered in the studies of health-promoting properties of diet [28,29,30].

## 3. Taste Impairments

Taste and olfactory impairments, collectively known as chemosensory disorders, are alterations of the normal gustatory and olfactory functioning that cause taste or smell losses or distortions [31]. Considering that flavor perception results from the interplay of taste, smell, and chemesthesis, gustatory impairments may prove difficult to disentangle from purely olfactory impairments. Indeed, complaints of decreased taste function or taste loss usually reflect a decrease in the ability to smell (hyposmia) or a complete loss of this function (anosmia). Evidence from specialized clinical studies shows that only around 5% of patients suffer from taste disorders, with the majority of such patients display altered odor perception [32]. At the same time, considering the ability of taste and smell apparatus to regenerate after damage, some recovery and compensation of function is generally possible [33]. This review mainly focuses on taste, and although olfaction and gustation are often grouped together clinically, it is important to highlight that they are biologically distinct with unique features and lead to potential clinical problems with different degrees of dysfunction [34]. Regarding taste, disorders are generally divided into qualitative and quantitative disturbances, although several other classifications have been adopted. Ageusia and hypogeusia are quantitative taste disturbances that result in a complete loss or partial reduction of the ability to taste, respectively. Both of these dysfunctions are generally rare, mostly because of the functional redundancy within the gustatory system. As a reference, in a study performed on 761 healthy subjects, hypogeusia was present in 5% of the participants, while none displayed complete ageusia [35]. At the other end of the spectrum, hypergeusia refers to an extreme acuteness of the sense of taste. Parageusia and phantogeusia are qualitative taste disorders that cause distorted or hallucinatory taste sensations in the presence, or absence, of external stimuli, respectively. Collectively, taste disorders are grouped under the general terminology of dysgeusia, a term that generally refers to any kind of impaired taste sensation [36]. Taste impairment is not a life-threatening condition but it can cause sufficient discomfort to lead to loss of appetite and changes in eating habits, with possible effects on health (Table 1).

## 4. Determinants of Taste Alterations 

### 4.1. Genetics

Genetics plays a role in affecting and altering taste perception in several different taste modalities. In more extreme cases, genetic disorders such as familial dysautonomia may cause complete or severe depletion of taste papillae, with a consequent reduction or loss of ability to perceive tastes [37].

At an individual level, genetic variation (known as polymorphism) within taste receptor genes has been shown to generate different tasting phenotypes, in some cases diminishing the ability of tasting, in other cases enhancing it. The most well-known example of such a trait is the case of phenylthiocarbamide (PTC), a compound not found in nature that tastes extremely bitter to some people (defined as “tasters”) but not bitter at all to others (“non-tasters”). The inability to taste PTC and structurally related molecules is inherited in a nearly Mendelian recessive manner, and >70% of the variation in perception is mediated by three single nucleotide polymorphisms in the TAS2R38 bitter receptor gene. A few additional variations occur in this gene in different worldwide populations [38,39], some of which influence the perception of PTC bitterness. Genetic variations also affect the perception of a bitter aftertaste of a number of widely used artificial sweeteners. These affect some individuals but not others. For example, polymorphisms in TAS2R43 and TAS2R44 affect the perception of a bitter aftertaste of saccharin and acesulfame K [66], while variants in TAS2R4 and TAS2R14 alter the bitter aftertaste of stevioside [40]. Over the past several years, a number of taste receptor gene polymorphisms have been identified, and their possible implication on eating behaviors and health (due to the extraoral expression on taste receptors) has been investigated [29].

Taste losses and gains can also be found across different species, and studies of taste receptor pseudogenes have illustrated a complex evolutionary history in vertebrates, made up of a succession of gene gain and loss processes [67]. Lineage-specific taste losses have indeed been discovered in several vertebrates’ evolutionary adaptations to dietary changes. A remarkable example is represented by the loss of sweet taste perception of cats and other felids, likely reflecting their high-protein diet, with little carbohydrates and no simple sugars [68].

### 4.2. External Factors

Although genetics plays a role in taste alterations, most such alterations arise due to environmental factors. Many chemicals (both from external sources—including exposure to toxic chemicals and to industrial agents—as well as internal sources including liver and kidney failure, uremia, and diabetes) have been shown to adversely affect taste perception by interfering with the chemical composition or quantity of saliva, changing the oral mucus membranes, reducing the number, activation, or the regenerative ability of taste receptor cells and efficacy of signal transduction, and directly affecting taste nerves [41].

A number of drugs (including some antimycotics, antibiotics, anti-inflammatories, immunosuppressants, neurologic medications, and psychiatric drugs) have been reported to adversely affect taste perception and eating behaviors. Among these, special attention is given to cancer chemotherapy medications due to their importance in terms of the number of treated patients [42,43].

Gustatory dysfunctions may also arise after trauma, as a result of direct injury to the tongue and to taste buds, damage to cranial nerves, brain contusion, or hemorrhage. Although such abnormalities usually occur immediately after injuries, in some cases they can arise several months after trauma [44].

Postoperative ageusia or dysgeusia, generally caused by partial or complete nerve transection, traction, or stretching, have also been described, in particular following removal of third molars, middle ear, head and neck surgeries, and after general anesthesia [45,46]. Similarly, radiation therapies in patients with head and neck cancers have been shown to significantly affect taste functions, by impairing intensity responsiveness, taste recognition, and detection thresholds because of radiation-induced atrophy of the taste buds [46]. Although some recovery of such impairments may occur in short term (within two or three months after the radiation treatments), a partial taste loss often persists in these patients for periods as long as twenty years [51]. Oral health has also been implicated in taste alterations. Appliances such as dentures, tooth prosthetics, and filling materials, as well as medical conditions such as lichen planus, burning mouth syndrome, and dry mouth are additional potential causes of hypogeusia, dysgeusia, or parageusia. These appear to be mainly the result of damage to central or peripheral nerves and/or decreased salivary flow rate [47,48].

Evidence shows that cigarette smoking may lead to alteration of taste buds and vascularization of the fungiform papillae, diminishing the ability to taste. Although in the scientific literature there is an inconsistent relationship between the use of nicotine products and taste dysfunctions, tobacco smoking and smoke inhalation have been linked to an increased risk of developing respiratory infections and dental problems, which may affect people’s ability to taste [49]. However, flavors play a role in compensating or masking consumers’ perceptions by introducing chemosensory sensations such as heat, tingling, burning, or cooling. Cigarettes and other tobacco products contain bitter compounds such as nicotine, which contribute to the chemosensory properties of tobacco, and bitter taste receptor gene variants have been shown to be variably associated with smoking status, depending on the populations studied [69]. Menthol is a flavoring additive commonly added in cigarettes and tobacco products to reduce and mask the harshness and bitterness of smoke due to its cooling and anesthetic properties, particularly used among African Americans [70].

High alcohol consumption can also compromise normal taste functions, mostly by changing the sensitivity of taste receptors and interfering with the absorption of micronutrients such as vitamins B and A, and the mineral zinc. These lead to functional changes in saliva and morphological changes in the taste buds, resulting in increased apoptosis in the salivary glands. Alcoholics have, in fact, been reported to show less sensitivity to sweet taste compared to control groups, suggesting that drinking habits may influence the choice of foods, with a greater preference for foods with higher sucrose concentration [50].

### 4.3. Nutritional Factors

Malnutrition and altered dietary intake have also been shown to produce dysgeusia or other forms of longstanding taste alterations. Evidence that nutritional deficiencies, either of primary or secondary origin, may cause taste perception alterations, or exacerbate the effects induced by aging, have been well documented. In particular, zinc plays an important role in various essential metabolic pathways and is a constituent of a large variety of metalloenzymes. It is a component of proteins involved in taste transduction such as gustin, and zinc deficiency has been identified as a causative factor in taste disorders that may lead to appetite changes [52].

In addition, specific foods have been shown to elicit taste disturbances. An interesting example is the case of pine mouth, otherwise known as Pine Nut Syndrome (PNS). PNS is uncommon dysgeusia that generally begins 12–48 h after consuming pine nuts mostly of a single species (*Pinus armandii*). It is characterized by a bitter-metallic taste, usually amplified by the consumption of other foods, lasting up to 4 weeks. Although no potential triggers or common underlying medical causes have been identified in subjects affected by this syndrome, it has been hypothesized that homozygous PTC taster status may be a potential contributor for pine mouth events [53].

Among other examples of natural compounds that can temporarily modify taste perception are miraculin and gymnemic acids. Miraculin is a protein isolated from the berries of *Richadella dulcifica*, which has no taste by itself at neutral pH but is able to convert sour stimuli to sweetness. This has resulted in the fruit of this plant being referred to as miracle fruit. Gymnemic acids are saponins isolated from the plant *Gymnema sylvestre* that are known to selectively suppress taste responses to various sweet compounds without affecting responses to salty, sour, and bitter substances. Their mechanisms of action on sweet taste receptors have been explained at the molecular level [71,72].

### 4.4. Biological Factors

Problems with taste perception increase with aging, in particular after the seventh and eighth decades. Age-related declines in both taste threshold sensitivity and the perceived intensity of suprathreshold tastes have been observed, in particular, for the sour and bitter taste, although other taste modalities seem to be affected [73].

More specifically, elderly people seem to be particularly prone to localized taste losses on specific areas of the tongue. Indeed, patients >65 years of age have been shown to be significantly more likely than young or middle-aged patients to experience phantogeusia and hypogeusia [74].

In addition, around 27 percent of adults ≥80 years and older report having had a problem with their sense of taste, including changes in taste sensation over time [31].

Even if these taste losses are more common among older people, the exact causes of these disturbances are not fully understood. Available evidence shows a reduction in the density of taste buds and papillae and reduced neural responsiveness to tastes with age [55]. In addition to these physiological changes associated with aging, the most common contributors to taste disorders in the elderly are those previously discussed such as drug use, zinc deficiency, and both oral and systemic diseases [36,75]. Moreover, neurological diseases typically manifested in the elderly, such as dementia and Parkinson syndrome, have been shown to be associated with reductions of gustatory abilities beyond that of normal aging, supposedly through the involvement of the frontal cortex, although secondary causes such as modifications in the salivary constitution may also play a role [56,57].

Taste preference, taste detection thresholds, and reactivity to taste stimuli are also known to be affected by sex differences. These are thought to occur at the taste-buds level. Behavioral and neurophysiological evidence indicates that such differences occur throughout the gustatory system, and that sex steroid hormones may modulate taste processing in the brain since receptors for sex hormones appear to be prominent in several nuclei associated with central gustatory pathways. In particular, estrogen levels can impact taste-elicited activity in the periphery and the brainstem, especially in the limbic pathway [54].

There remains a need to fully characterize the variation in the gustatory function that occurs in pregnancy, and it has been hypothesized that taste changes during pregnancy are not only due to changes in progesterone and estrogen but are likely affected by other factors, such as other endocrine factors, altered immune responses, or changes in oral health [76].

### 4.5. Viral Illness

Viral upper respiratory and influenza infections are important causes of losses of taste and smell, mostly due to the nasal blockage, obstruction, and swelling of the mucosa generated by increased mucus production and changes in mucus composition [58,59]. It has also been shown that inflammation responses in taste tissues play a role in the pathogenesis of taste dysfunction and cell turnover in taste buds [60,61]. In most cases, taste and smell losses are temporary and they resolve once the cold and influenza symptoms disappear. Some patients who have experienced viral illnesses, however, develop taste dysfunctions characterized by a complete loss or a reduced ability to detect and recognize taste stimuli [77,78]. Such events do not always have an acute onset. They can also develop gradually over time, generally triggered by an upper respiratory viral infection, oral cavity infections, or viral hepatitis.

Among these, taste disturbances have been shown to develop in patients who have been affected by infectious diseases caused by viruses responsible for the common cold (Rhinoviruses), influenza (Orthomyxoviridae), or hepatitis (Hepatoviruses), among others. In total, there are more than 200 viruses causing upper respiratory tract infections that can lead to a decrease in chemosensation [79]. Disturbances or loss of smell, taste, and chemesthesis have also emerged as predominant symptoms of the recent Coronavirus disease 2019 (COVID-19), caused by Severe Acute Respiratory Syndrome coronavirus strain 2 (SARS-CoV-2), as well as of previous endemic and pandemic coronaviruses such as Middle East Respiratory Syndrome Coronavirus (MERS-CoV) and SARS-CoV [61,62,80]. In a striking preliminary statistic, up to 80% of subjects affected by COVID-19 have reported taste disturbances such as dysgeusia, ageusia, or olfactory-related changes such as hyposmia or anosmia [61]. As previously noted, self-reported taste and smell disturbances are not always reliable, as reported taste losses may actually be a conflation with changes in olfaction. To ensure that the reported disturbances are properly ascertained and attributed to the correct sense, it is important to discern taste from flavor when self-reporting their taste/olfactory status. Recent studies have sought to accomplish this by implementing a check all that apply (CATA) question in questionnaires so that participants are given the option to report changes in specific taste qualities (i.e., salty, sour, sweet, bitter, or umami/savory) that they feel have been affected [81]. Another complication arises due to the fact that taste-related disturbances may be due to drugs prescribed for this viral illness, rather than from the actual infection [62]. However, recent evidence suggests that chemosensory disturbances induced by SARS-CoV-2 may be affected through mechanisms distinct from those employed by other coronaviruses, influenza, or rhinoviruses [61,81,82]. Indeed, compared to acute cold patients, taste functions are significantly worse in people affected by COVID-19: in fact, not only global but sweet and, in particular, bitter gustatory scores have been shown to discriminate well between COVID-19 and acute cold patients [63].

These results suggest that taste disturbances reported by COVID patients may be the result of an actual impairment of gustatory abilities, rather than simply an olfactory dysfunction. With our current limited understanding of SARS-CoV-2, studies are ongoing to better elucidate the pathogenesis of the symptoms associated with this virus, and how SARS-CoV-2 influences smell, taste and chemesthesis remains uncertain. Recent studies, however, show that SARS-CoV-2 appears to infect cells through interactions between its S-protein and the angiotensin-converting enzyme 2 (ACE2) receptor on target cells to gain entry to cells via binding with the viral spike protein [61]. The SARS-CoV-2-mediated infection also requires a subsequent cleavage of this S protein, likely by the host cell serine protease TMPRSS2 and other proteases [83]. In addition, it has also been suggested that GPCRs expressed on the epithelial cells of the alveoli of the lungs may play a role in the cellular entry pathway for the related SARS-CoV-2 virus. This has led to postulations that olfactory, umami, sweet, and bitter receptors, being GPCRs, may also be additional targets of the virus [63,84]. This hypothesis, although of relevance and interest, awaits experimental confirmation, especially considering the large number of different GPCRs that are ubiquitously expressed. ACE2 and TMPRSS2 are expressed in multiple human organ systems including the nasal respiratory epithelium and the olfactory neuroepithelium [85,86]. Indeed, a recent analysis of more than 80,000 human genomes suggests possible associations between genetic polymorphisms in ACE2 and TMPRSS2 with COVID-19 susceptibility, severity, and clinical outcomes [87]. Studies of sour, bitter, and sweet/umami taste receptor cells have been shown to express ACE2 but little to no TMPRSS2: they do, however, show a strong expression of cathepsins B and L that may operate as proteases to cleave SARS-CoV-2 S protein [61]. A possible mechanism of taste and chemesthetic COVID-19-related disturbances could, therefore, result from the direct infection of cells in the tongue, in addition to secondary consequences of obstruction due to inflammation, or damage to cranial nerves (i.e., trigeminal, facial, and glossopharyngeal) following the release of inflammatory cytokines that may be responsible for damaging the taste receptors and altering their transduction or expression [61,62,63,64]. It has also been hypothesized that such immune responses to SARS-CoV-2 viral replication may lead to changes in localized cellular zinc homeostasis in oral gustatory cells that may also be accompanied by hypozincemia, resulting in dysgeusia [65]. Such proposed mechanisms, although plausible and based on the interpretation of the limited current evidence, still lack confirmation from direct evidence: a further understanding of dysgeusia in relation to COVID-19, therefore, still needs additional mechanism-based information.

## 5. Potential Implications on Dietary Habits

Taste or smell losses or disturbances pose risks on multiple levels and have the potential of significantly lowering the quality of life. Potential exposure to toxins, for instance, is increased in people with smell and taste impairments as they may have difficulty or be unable to detect the odor or taste of rotting or spoiled food. Smell impairments significantly enhance, for example, the risk of missing the chemical warning signs of a gas leak or the presence of poisonous fumes, putting people at potential severe risks [88]. A diminished or lost sense of taste makes it more difficult for people to appreciate and enjoy eating, causing them to avoid many foods. As a result, alterations of smell and taste have been associated with depression and anxiety [89,90]. Persistent dysgeusia, therefore, has the ability to lead to serious complications in terms of health consequences, generated from shifts towards unhealthy eating habits. For instance, a distorted sense of taste in individuals that are required to maintain a specific diet for health-related purposes, such as diabetics, or individuals with celiac disease or high blood pressure, may result in a change of their eating habits, raising the risk of metabolic and cardiovascular diseases [91]. Taste impairments have proven to contribute to undernutrition and malnutrition of cancer patients, who often have lost body weight already at the time of the diagnosis, further deteriorating their nutritional status and life quality [42]. Similarly, a diminished threshold for salt perception may cause people to increase their discretionary salt intake to improve food palatability, increasing their risk of cardiovascular disease [91,92]. For these reasons, guidelines that take into account food characteristics such as colors, texture, packaging, labeling, in addition to flavor and nutrients, have been proposed to food science communities to design special foods for consumers affected by smell and taste losses in an attempt to encourage patients with such disturbances to eat well while creating new business opportunities for food companies [93].

As discussed, a high number of subjects affected by SARS-CoV-2 have reported taste disturbances. To our knowledge, the repercussions on dietary changes of such disturbances have not yet been reported. This is because these are usually long-term effects that need time to be recorded and assessed. In addition, these disorders may have been confounded with the changes in dietary habits that the recent COVID-19 pandemic event has caused. Indeed, following the changes in livelihoods, social life, and lifestyles (such as quarantine, work and school closures, and travel restrictions), people’s behaviors have changed, particularly regarding the food they eat or how they prepare it. During the pandemic, in fact, direct access to fresh foods may have been more limited as a result of food poverty and insecurity, blockages of the supply and production chains, or even panic buying, inducing people to increase their consumption of more processed foods and products with longer shelf-lives [94]. Regarding Italy, for example, a recent study found that 86% of respondents reported that they were unable to sufficiently control their diet due to isolation, lack of stimuli, boredom, and changing food routines: they were thus being inclined to increase food intake to feel better. The authors further speculate that this may have resulted in an increase of overall caloric intake because of the amount and type of food consumed daily, likely caused by the sudden availability of having more time to cook and consume meals. This may have compounded by the fact that, at least for the quarantine period, the only freedom allowed was to go grocery shopping, as reflected by an increase of home cooking and artisanal food production [95,96]. A negative effect of home confinement on physical activity, food consumption, and meal patterns was emerging also from an international survey [97]. At the same time, recent views have pointed out how nutrition may influence COVID-19 susceptibility and have long-term consequences, influencing both the host’s response and the pathogen’s virulence [98]. Given this, current nutrition advice that may have repercussions on current and future dietary habits, and may see a recommended increase in consumption of dietary fiber, whole-grains, micronutrients, and antioxidants to boost immune function, with potential implications not only for consumers but also for the food industry that will have to be able to adapt and cope with such future changes [95,99]. The current situation has clearly challenged our food systems and food security paradigms, creating the basis for a new era, providing both opportunities and challenges for food manufactures, through the increase of digital technologies and unconventional delivery systems such as online ordering, delivery, and smartphone apps, among others [96].

## 6. Conclusions

Taste has a major role in individuals’ everyday lives, acting as a gatekeeper of food ingestion. On one hand, bitter, salty, and sour tastes are thought to have come about as a defensive and protective mechanism to prevent the ingestion of potentially toxic foods and to assure internal sodium or acid–base balance. On the other hand, sweet foods are naturally attractive and accepted since carbohydrates serve as the main energy source for animals. Lastly, umami senses amino acids in proteins, which naturally occur in meats, vegetables, and fermented products and are essential for humans. Impairments of such normal gustatory functioning cause taste losses or taste distortions and are caused by a variety of factors including genetics, nutrition, biology, external factors, and viral illness. Taste dysfunction, and chemosensory disfunction in general, have also emerged as prominent symptoms of SARS-CoV-2 infection in the current COVID-19 pandemic. Taste impairments are not considered to be life-threatening conditions, and such alterations, reductions, or losses of any of these taste qualities have often been considered as secondary or less important problems. This, however, does not take into account their profound impacts on individuals’ dietary habits and overall quality of life, which in turn have possible effects on health. Taste disturbances should, therefore, be given more consideration than they have received, and understanding the mechanisms that underlie these disturbances in COVID-19 patients may provide additional important insights into the molecular pathophysiology associated with this disease.

## Figures and Tables

**Figure 1 nutrients-12-03284-f001:**
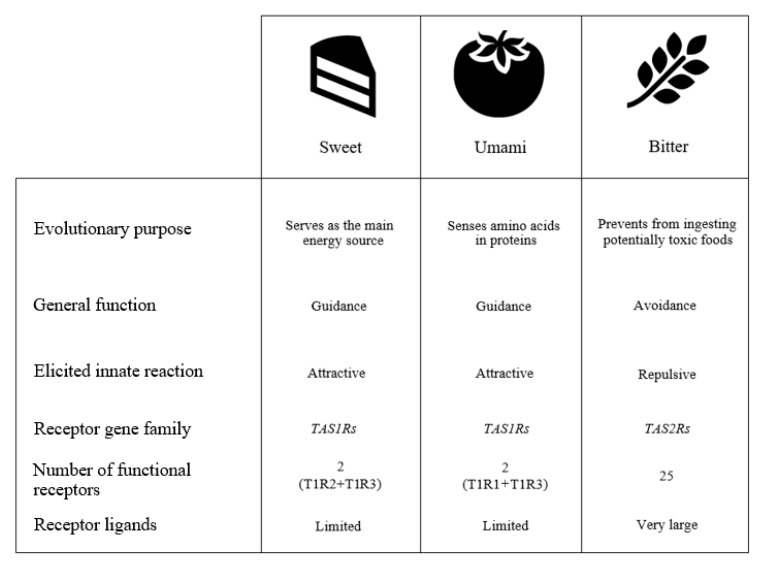
Overview of sweet, umami, and bitter tastes characteristics.

**Table 1 nutrients-12-03284-t001:** Overview of main determinants of taste alterations and their causes.

Factor	Sub-Factor	Proposed Mechanisms of Action	References
Genetics	Genetic disorders (e.g., Familial dysautonomia)	Complete or severe depletion of taste papillae	[37]
	Single nucleotide polymorphisms on taste receptor genes	Influence individuals’ ligand-receptor binding, generating different tasting phenotypes	[38,39,40]
External	Chemicals (both from external and internal sources)	Interfere with the chemical composition or quantity of saliva; change the oral mucus membranes; impair taste receptor cells and efficacy of transduction; directly affect taste nerves	[41]
	Drugs (e.g., antimycotics, antibiotics, anti-inflammatories, immunosuppressants, neurologic medications)	Changes in saliva production, secretion, quantity, and diffusion; damage to cranial nerves; modification of afferent pathways from the central nervous system	[41,42,43]
	Trauma (e.g., brain contusion, hemorrhage)	Direct injury to the tongue and to taste buds; damage taste nerves	[44]
	Surgeries (e.g., removal of third molars, middle ear, head and neck surgeries, general anesthesia)	Partial or complete nerve transection, traction, or stretching	[45,46]
	Oral appliances and conditions	Damage to central or peripheral nerves; decreased salivary flow rate	[47,48]
	Smoking	Increases respiratory infections and dental problems	[49]
	Alcohol consumption	Changes the sensitivity of taste receptors and interferes with the absorption of micronutrients, leading to functional changes in saliva and morphological changes in the taste buds	[50]
	Radiation therapies (e.g., in patients with head and neck cancers)	Impair intensity responsiveness, taste recognition and detection thresholds	[51]
Nutrition	Zinc deficiency	Is a component of proteins involved in taste transduction	[52]
	Specific foods (e.g., *Pinus armandii* pine nuts)	No potential triggers or common underlying medical causes have been identified yet	[53]
Biology	Sex	Sex steroid hormones may modulate taste processing in the brain	[54]
	Aging	Reduction in taste buds and papillae density; reduced neural responsiveness to tastes	[55]
	Neurological diseases (e.g., dementia, Parkinson syndrome)	Involvement of the frontal cortex; changes in salivary constitution	[56,57]
Viral illness	Common cold (Rhinoviruses), Influenza (Orthomyxoviridae), MERS (MERS-CoV) Hepatitis (Hepatoviruses)	Mostly due to the nasal blockage, obstruction and swelling of the mucosa generated by increased mucus production, and changes in mucus composition; generate inflammation responses in taste tissues	[58,59,60,61]
	COVID-19 (SARS-CoV-2)	Appears to impair more sweet and bitter tastes; may result from direct infection of cells in the tongue; secondary consequences of obstruction due to inflammation; damage to cranial nerves following the release of inflammatory cytokines; and/or lead to changes in localized cellular zinc homeostasis in oral gustatory cells	[61,62,63,64,65]

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
