# Peer review of "Alteration, Reduction and Taste Loss: Main Causes and Potential Implications on Dietary Habits"

_nutrients, 2020, doi:10.3390/nu12113284_

Round 1

Reviewer 1 Report

This is an interesting review about taste alterations and loss. The authors described main causes and potential implications on dietary habits. Moreover, they focused on taste disturbances due to recent SARS-CoV2 pandemic. The paper is well written.

The authors adequately addressed normal anatomy and physiology before describing taste disorders and their causes.

The authors should include head and neck surgery and radiation therapy for cancer among external causes of taste alterations and discuss them. Moreover, neurological disorders that affect taste should be added.

I think that some tables and/or graphics summarizing information may help the readers (e.g. taste receptors; main causes of dysgeusia and their action).

Author Response

We thank Reviewer 1 for the positive and constructive feedback on our manuscript. We particularly appreciated the valuable suggestion to include some tables and/or figures summarizing the discussed information to increase the readability of the manuscript.

Following this advice, we now have included on page 4, Figure 1 illustrating the three analyzed tastes, pointing out their receptors and main differences. In addition, on pages 5-6, we have now added Table 1 providing the reader with an overview summary of the main causes of dysgeusia and their mechanisms of action discussed in chapter 4.

As suggested, we have now included in both in Table 1 and on page 7 lines 237-243, head and neck surgery and radiation therapy for cancer among the external causes of taste alterations as follows: “ Postoperative ageusia or dysgeusia, generally caused by partial or complete nerve transection, traction or stretching, have also been described, in particular following removal of third molars, middle ear, head and neck surgeries, and after general anesthesia [48,49]. Similarly, radiation therapies in patients with head and neck cancers have been shown to significantly affect taste functions, by impairing intensity responsiveness, taste recognition and detection thresholds because of radiation-induced atrophy of the taste buds [49]. Although some recovery of such impairments may occur short term (within two or three months after the radiation treatments), a partial taste loss often persists in these patients for periods as long as twenty years [50].

As also suggested, we have further added a discussion of neurological disorders that affect taste both in Table 1 and on page 9 lines 305-307 as follows: “Moreover, neurological diseases typically manifested in the elderly, such as dementia and Parkinson syndrome, have been shown to be associated with reductions of gustatory abilities beyond that of normal aging, supposedly through the involvement of the frontal cortex, although secondary causes such as modifications in salivary constitution may also play a role [66,67].

Reviewer 2 Report

Please find below my comments regarding the manuscript entitled "Alteration, Reduction and Taste Loss: Main Causes 2 and Potential Implications on Dietary Habits."

In general, the information was very comprehensive and well-compiled. However, the information in the review has largely been reviewed elsewhere. Section 2 of the manuscript, "How we taste", has been reviewed elsewhere several times and can be cited. Further, a review entitled "Dysgeusia in COVID-19: Possible Mechanisms and Implications" by Lozada-Nur et al. (June 2020) explains the likely mechanisms by which taste dysfunction develops in SARS-CoV-2 infection. Finally, the portion of the review from line 382-407 seems out of scope and belonging to a different review.

Line 182: "Genetics may play a role...". Not sure if using the word "may" was intentional, but would suggest removing. The role of genetics in taste is a question of how, not if.

Section 4.5 can be broken into paragraphs for improved readability. 

Lines 321-322: "It is therefore essential to ensure that patients are properly discerning taste from flavor when self-reporting their taste/olfactory status." Why? This requires elaboration. What are the consequences for patients who do not make this distinction?

Lines 341-344: "In addition, it was also suggested that GPCR expressed on the epithelial cells of the alveoli of the lungs may be involved in the pathway of SARS-CoV-2 cells entry: such hypothesis is of relevance and interest, considering that olfactory, bitter, and sweet and bitter receptors are GPCRs, and suggests that they may also be possible targets of the virus [75,77]." Considering the vast number of different GPCRs expressed in various cell types, this statement seems like a stretch. 

Lines 351-352: "A possible mechanism of taste and chemesthetic COVID-19-related disturbances could therefore result from the direct infection of cells in the tongue [...]". Without providing any evidence that the virus can enter a taste receptor cell, as mentioned in the previous comment, this mechanism does not yet have plausibility. While saying that the mechanism is "possible" is accurate, this statement is misleading if not accompanied by another statement of the limitations of this proposed mechanism given current evidence.

Section 6: The conclusion is too focused on basic taste information that has not only been covered in other reviews, but is not the focus of this particular review. Suggest revising the conclusion to include more information regarding taste loss, rather than only including one sentence at the end.

Author Response

We thank Reviewer 2 for the positive general comment and for providing such a detailed review and analysis of our manuscript. We are also grateful for the many comments that surely contributed to improve not only the content but also the readability of our paper. Please find below our comments and considerations, point-by-point:

Comment: “However, the information in the review has largely been reviewed elsewhere. Section 2 of the manuscript, "How we taste", has been reviewed elsewhere several times and can be cited. Further, a review entitled "Dysgeusia in COVID-19: Possible Mechanisms and Implications" by Lozada-Nur et al. (June 2020) explains the likely mechanisms by which taste dysfunction develops in SARS-CoV-2 infection. Finally, the portion of the review from line 382-407 seems out of scope and belonging to a different review.”

Response: It is true that many reviews have already addressed both the general topics of taste perception and transduction, taste receptor genes, and taste disturbances. However, regarding the latter, we found that most of these focused on the general mechanisms of such disturbances, without elaborating or linking them to their possible implications on food habits, which may be significant. Similarly, recent reports addressing the mechanisms by which taste dysfunction develops in SARS-CoV-2 infection, to our knowledge, have not provided insights on the repercussions of this on dietary habits, nor have they linked them to other causes of taste disturbances. The effects of COVID-19 on individuals’ dietary habits have indeed been mostly discussed in relation to the lockdown/quarantine measures.

The aims of our Review are to give a more up-to-date overview of taste perception mechanisms, to provide a comprehensive list of the main causes of taste disturbances, to discuss such in relation to their potential implications on dietary habits, and finally to contextualize these in light of the current COVID-19 pandemic.

To better convey this and to address this point by Reviewer 2, we have now modified our manuscript as follows:

- page 2, line 89. We have added another reference, citing a previous well-known review focusing on taste perception and we have modified the text as follows: “There are two types of transmembrane receptors relevant for taste: ion channels, which mediate sour (H+) and salty (Na+) tastes, and G protein coupled receptors (GPCRs), which mediate sweet, umami and bitter tastes, as well described in previous reviews [14,15].

- page 3, lines 104-107. We have added more references to previous reviews focusing on taste receptor genes and their variants and modified the text as follows: “There is more information and scientific certainty about the nature of the receptors for sweet, umami and bitter tastes, which belong to the GPCR class. Given that the overall number of sweet, umami, and bitter taste receptors genes and their known variants is large, and that many reviews have already covered this topic [14,15,20], we here provide a short introduction to this topic to provide general background for the reader, and provide context on taste disturbances in the current COVID-19 pandemic”

- page 10, lines 375-378. We thank Reviewer 2 for pointing out the reference by Lozada-Nur et al. This report provides an additional mechanism by which taste dysfunction may develop in SARS-CoV-2 infection that we had not included. We note that although Lozada-Nur and colleagues reviewed and discussed the possible mechanisms and implications of dysgeusia in COVID-19, and also highlighted the potential genes involved, their letter did not address the potential effects of such alterations on dietary habits, nor did they point out possibile genetic susceptibilities conferred by such genes. One paper highlights this and, for instance, (Hou et al., reference 80 of our review) has been published more recently (July 2020). As suggested by this reviewer, we have now included this reference and amended the manuscript as follows: “It has also been hypothesized that such immune responses to SARS-CoV-2 viral replication may lead to changes in localized cellular zinc homeostasis in oral gustatory cells that may also be accompanied by hypozincemia, resulting in dysgeusia [88].”

- pages 10-11, lines 382-407. We respectfully disagree with Reviewer 2 on the statement that this part of the manuscript is out of scope and seems to belong to a different review. On the contrary, we believe we’ve clearly illustrated the profound impacts that taste/smell losses or disturbances may have on peoples’ dietary habits and their overall quality of life, with possible effects on health. We feel this is an important part of our manuscript and that it serves to raise awareness of the importance of taste and smell disfunctions that are generally not attention they deserve simply because they are not viewed as life-threatening conditions.

- page 6, line 193. "Genetics may play a role...". We have now removed “may” as suggested.

- page 9, section 4.5. We have now broken this section into three paragraphs to improve readability.

- page 9, Lines 343-343. The sentence "It is therefore essential to ensure that patients are properly discerning taste from flavor when self-reporting their taste/olfactory status" is used to note that flavor perception results from the interplay of taste, smell and chemesthesis, gustatory impairments. These often prove difficult to disentangle from purely olfactory impairments. We were not addressing consequences for patients who do not make this distinction. To better clarify this, we have now modified the text as follows: “To ensure that the reported disturbances are properly ascertained and attributed to the correct chemical sense, it is important to discern taste from flavor when self-reporting their taste/olfactory status”.

- page 10, lines 363-367. We agree with Reviewer 2 that, considering the vast number of different GPCRs expressed in various cell types, it may be an overgeneralization to say “..olfactory, bitter, and sweet and bitter receptors are GPCRs, and suggests that they may also be possible targets of the virus”. However, we note that recent reports including Singh et al. (Apr 2020) and Huart et al. (July 2020) have postulated such possibility. To better address this point, we have now modified the text as follows: “In addition, it has also been suggested that GPCRs expressed on the epithelial cells of the alveoli of the lungs may play a role in the cellular entry pathway for the related SARS-CoV-2 virus. This has led to postulations that olfactory, bitter, and sweet and umami receptors, being GPCRs, may also be additional targets of the virus [81,83]. This hypothesis, although of relevance and interest, awaits experimental confirmation, especially considering the large number of different GPCRs that are ubiquitously expressed.”

- page 10, lines 376-386. In order to empower the reader to make their own critical judgement, we have now added an additional sentence to the text, highlighting how the proposed mechanisms for taste disturbances associated with COVID-19 still need confirmation from direct evidence; “Such proposed mechanisms, although plausible and based on the interpretation of the limited current evidence, still lack confirmation from direct evidence. A further understanding of dysgeusia in relation to COVID-19 therefore still needs additional mechanism-based information.

- page 11, Conclusions. We agree with Reviewer 2 that our original conclusion may have been too focused on basic taste information and less focused on taste loss. To address this, we have now amended the text as follows: “Taste has a major role in individuals’ everyday lives, acting as a gatekeeper of food ingestion. On one hand, bitter, salty and sour taste are thought to have come about as a defensive and protective mechanism to prevent ingestion of potentially toxic foods and to assure internal sodium or acid-base balance. On the other hand, sweet foods are naturally attractive and accepted, since carbohydrates serve as the main energy source for animals. Lastly, umami senses amino acids in proteins, which naturally occur in meats, vegetables and fermented products, and are essential for humans. Impairments of such normal gustatory functioning cause taste losses or taste distortions and are caused by a variety of factors including genetics, nutrition, biology, external factors and viral illness. Taste disfunction, and chemosensory disfunction in general have also emerged as prominent symptoms of SARS-CoV-2 infection in the current COVID-19 pandemic. Taste impairments are not considered to be life-threatening conditions, and such alterations, reductions, or losses of any of these taste qualities have often being considered as secondary or less important problems. This, however, does not take into account their profound impacts on individuals’ dietary habits and overall quality of life, which in turn have possible effects on health. Taste disturbances should therefore be given more consideration than they have received and understanding the mechanisms that underlie these disturbances in Covid-19 patients may provide additional important insights into the molecular pathophysiology associated with this disease.”

Round 2

Reviewer 2 Report

none